# The association of motivation with mind wandering in trait and state levels

Toshikazu Kawagoe[1]*, Keiichi Onoda[2], Shuhei Yamaguchi[3]

**1** College of Contemporary Psychology, Rikkyo University, Niiza city, Saitama, Japan, **2** Faculty of Psychology, Otemon Gakuin University, Ibaraki city, Osaka, Japan, **3** Faculty of Medicine, Shimane University, Izumo city, Shimane, Japan

* toshikazukawagoe@gmail.com

**Data Availability Statement:** Data cannot be shared publicly because of the ethical policy because we did not explicitly denote that the data will be openly available in publication during informed consent. However, data are available

## Abstract

Mind wandering (MW) is a phenomenon in which attention drifts away from task-related thoughts toward task-unrelated thoughts. Recent studies have demonstrated that MW occurs during tasks in which participants are unmotivated. However, motivation ranges on a continuum from trait to state. We examined the association between trait-state motivation and trait-state MW. Participants (176 undergraduate students 18–24 years old; 68 male) completed three questionnaires for our trait level investigation. State level indices were measured using the experience sampling method with 104 students completing a sustained attention to response task. Through correlation analyses, we demonstrated an association between motivation and MW within the same dimension (trait and state, respectively) but found no association across dimensions in which the correlation coefficient was nearly zero. We show the significant association between motivation and MW whose novelty is especially evident in the trait level. Although the relationship between motivation and MW is substantial, trait-state dimensionality would be important for them. The state MW is a phasic phenomenon driven by a range of factors, one being state motivation. The causality and confounding factors remain to be further studied.

## Introduction

Experiencing our minds drift away from tasks, especially undemanding, trivial ones, toward unrelated inner thoughts, fantasies, and other musings is a common occurrence for as much as 50% of our waking hours [1]. This is known as mind wandering (MW), which can be experienced in various situations and are often unintended and occurring beyond awareness [2–4]. To be accurate, MW is the umbrella term for the psychological phenomenon to which we refer, including *task-unrelated thought*, *stimulus-independent thought*, *self-generative thought*, and *zoning/tuning out*. Although the specific theoretical differences between these terms have been discussed [4,5], here we use the term "MW" to encompass the above phenomena following one of the most influential reviews in this field [2]. While MW can have some positive impacts (e.g., autobiographical planning, creative thinking, and attention cycling), it can also cause disruption of performance at various levels of the tasks at hand [3].

from the corresponding author upon reasonable request. The institutional point of contact for this study is Faculty Ethical Committee of contemporary psychology in Rikkyo University. Contact information (Email address) is ccp-rinri@ml.rikkyo.ac.jp.

**Funding:** Preparation of this manuscript was supported by Grant 19K14481 from Japan Society for the Promotion of Science (KAKENHI) to TK. The funder had no role in study design, data collection and analysis, decision to publish, or preparation of the manuscript.

**Competing interests:** The authors have declared that no competing interests exist.

Studies have indicated that motivation toward a task and MW while executing the task are significantly associated [6]. Participants with low motivation toward the ongoing task tend to experience more MW during the task, and this increase in MW is associated with large decrements in task performance. The association between state motivation and MW during an ongoing task is intuitive, as is the association of temporal psychological and physiological states with MW [7,8]. However, the trait-state dimension of motivation and/or MW has not been studied thoroughly. States are in-the-moment reactions to internal or external stimuli or situations, whereas traits that influence people's reactions or behaviors are more inherent to a person's character or personality. Certainly, people who are always unmotivated are defined as "apathetic" in the clinical population. Apathy has been defined as a "lack of motivation not attributable to diminished level of consciousness, cognitive impairment, or emotional distress" [9], and this homogeneous symptom has also been observed among the healthy population [10,11] as it has been suggested that apathy may be caused by diverse circumstances arising as part of normal development and experience [9]. Such "trait" motivation might affect "state" motivation, which would in turn affect task performance [12]. Additionally, MW has been assessed in terms of the trait-state dimension [13]. "Trait MW" is defined as how people perceive their level of MW in daily life, while "state MW" is determined by how people respond to thought probes requesting feedback on their momentary psychological experiences in the laboratory.

Uncovering the effects of trait motivation on MW could suggest how the trait aspect (i.e., individual characteristic) impacts MW at the trait level and state level (i.e., during the task at hand). In the present study, the trait level association between motivation and MW was assessed through questionnaires as described below. After confirming that association exists with a relatively larger sample, we conducted further investigation into the state level indices of motivation and MW. Trait and state MW are interconnected [13] and could affect ongoing task performance at the state level [6]. Here, we aim to replicate such associations, and to extend the results by including trait motivation and trait MW as factors.

## Methods

### Participants

A priori, we recruited as many participants as possible during the academic term. We recruited 185 undergraduate students for the questionnaire survey. Nine participants were excluded because of diagnosed psychiatric disorders ($n = 4$) and incomplete questionnaires ($n = 5$), resulting in a total of 176 participants (age: 20.5 years [SD: 2.3]; 68 male) analyzed for trait level characteristics. Of those 176 participants, 109 who agreed to participate in the detailed experiment underwent the state level investigation during an independent session. Two participants were excluded from the detailed investigation because they could not complete the task. Therefore, 107 participants (age: 20.9 years [SD: 2.8]; 36 male) were analyzed for the state level investigation. We considered the sample size to be relatively small but sufficient for this study because a minimum sample size of 84 is required to achieve "sizeable" correlation coefficients ($>0.3$) with a power of 0.8 [14]. The participants were healthy and none were undergoing neurological or psychiatric treatment. The Rikkyo University's ethics committee approved the study, which was conducted in accordance with the Declaration of Helsinki (1975, as revised in 2008) and the regulations of the Japanese Ministry of Health, Labour and Welfare. Furthermore, informed consent was obtained from all individual adult participants included in the study. All procedures performed in this study were done so in accordance with the ethical standards of the institutional research committee and with the 1964 Helsinki Declaration and its later amendments or comparable ethical standards.

## Questionnaires for trait level investigation

Three questionnaires were used in this part of the study: the apathy scale (AS) [15], the mind wandering questionnaire (MWQ) [16], and the daydreaming frequency scale (DDFS) [17]. The AS was conducted to assess the persistent characteristic of motivation, with higher scores indicating lower motivation. Although the AS was developed for apathy, which is a clinical level of amotivation, such questionnaire could be applied to a healthy population with a sufficient dispersion [11]. The Japanese version of AS which is a fourteen item 4-point scale whose scores range from 0 to 42 (e.g., "Are you interested in learning new things?"), was well known in Japan [18]. The score for AS was inverted in this study (Inv-AS) in which a higher score represents greater motivation. The MWQ and DDFS were conducted to assess the persistent propensity of MW tendencies. A difference between the MWQ (five items with a 6-point scale whose scores range from 5 to 30) and DDFS (twelve items with a 5-point scale whose scores range from 0 to 48) is that the MWQ predominantly taps into task-unrelated thought (e.g., "While reading, I find I haven't been thinking about the text, having, therefore, to read it again"), while the DDFS focuses especially on stimulus-independent thought (e.g., "On a long bus, train, or airplane ride, I daydream. . ."). These two have been frequently used to investigate MW tendency. In order to strengthen the reliability of expected results, we used both tasks given the results of a previous study that indicated a significant correlation between the two measurements [16]. The validated Japanese translated versions of MWQ and DDFS were used in this study [19]. There is no conceptual overlap between the constructs assessed by AS and by MWQ/DDFS.

**Behavioral task and questionnaire for state level investigation.** We conducted experience sampling using a sustained attention to response task (SART) to index MW at the state level. Experience sampling enables online assessment of momentary changes in the content of consciousness by collecting self-reports during the task [1,20,21]. The SART paradigm was administered by presenting digit stimuli from "1" to "9" on the screen in random order (Fig 1). The stimuli appeared in a square (3° of visual angle) presented at the center of the screen throughout the experiment; these stimuli were black and placed against a white background. The stimuli lasted one second, and the inter stimuli interval varied from one to three seconds. Participants were required to press a specified key on the laptop keyboard as soon as they detected the stimulus for all digits except "4" (i.e., non-targets). When participants pressed the key, the presenting digit disappeared. For the digit "4" (i.e., the target), participants had to restrain themselves from pressing a key and wait for the next stimulus. They were instructed to respond as fast and accurately as possible. At times, a thought probe was posed: "To what extent have you experienced task-unrelated thoughts in the moment just preceding the time of this thought probe?". Participants responded using a 7-point Likert scale as part of the "task-unrelated thought" question, where 1 = weak, indicating they were focused on the task, and 7 = strong, indicating their mind wandered completely. We set the answers as an index of MW. This index is hereafter referred to as "Probe." In this task, targets and thought probes were rarely presented. The target and the thought probe randomly appeared at the rate of 5% for all trials and of 2.3% (i.e., 15 probes / 650 presentations), respectively. Intervals of at least 30 stimuli occurred between each thought probe. This task took each participant about 25–30 minutes. To assess the temporal state of motivation, a modified intrinsic motivation inventory (IMI) [22] was used. The current IMI asks participants to rate their level of motivation for carrying out a given task at hand with nine items and consisting of a 7-point scale with scores ranging from 9 to 63 (e.g., "I put a lot of effort into this"). In the present study, we conducted this questionnaire after the SART.

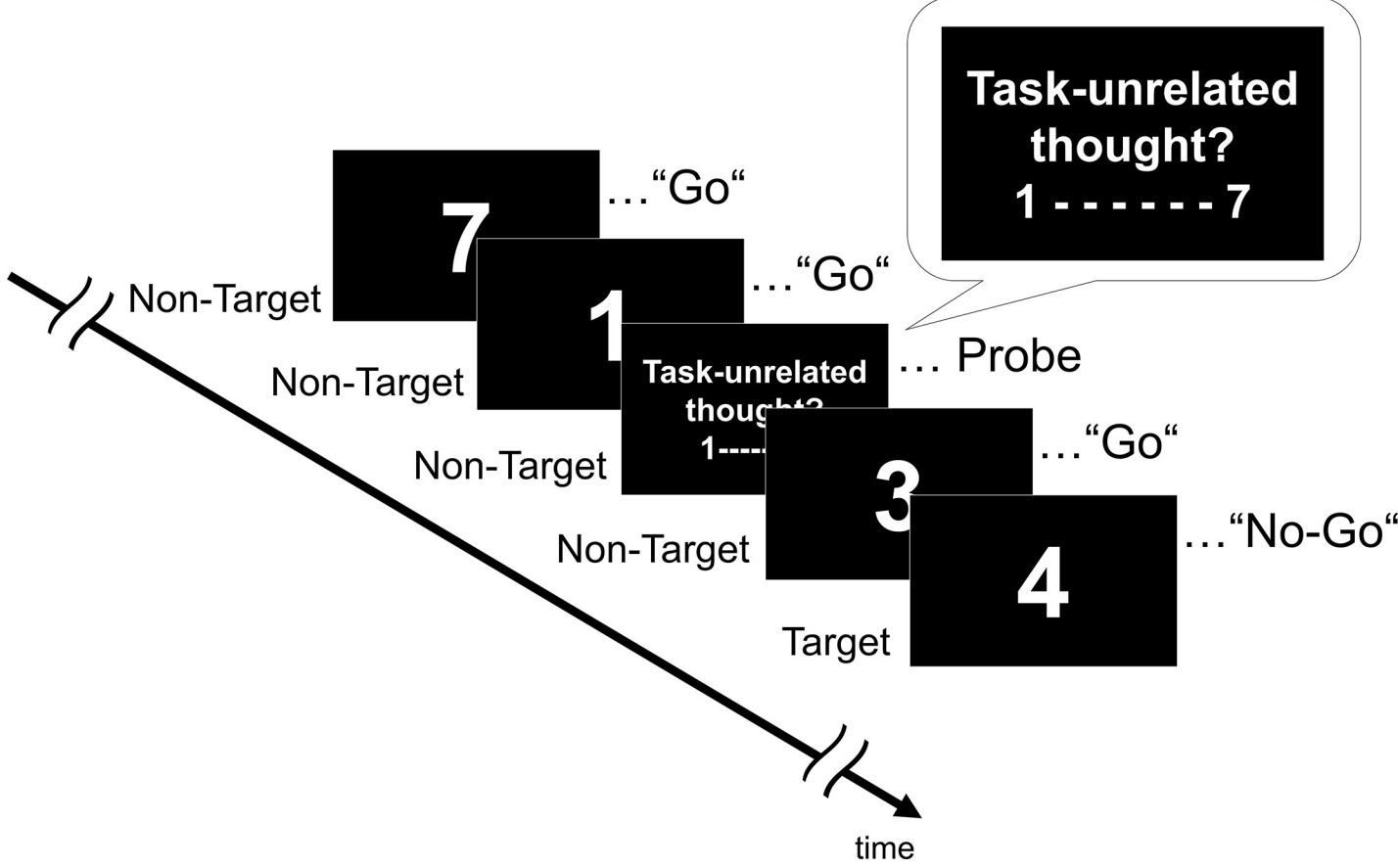

**Fig 1. Illustration of experience sampling with sustained attention response task.** Participants were presented with target (appearing at a rate of 5%) and non-target stimuli and had to press a key to indicate that the presenting stimulus was not a target (i.e., 4). The probe appeared at a rate of 2% (i.e., 15 probes) of the whole presentation, and asked the participants to what extent they experienced task-unrelated thoughts in the moment preceding this thought probe. The language in this figure was abbreviated for illustrative purposes.

## Results

### Descriptive statistics

As presented in the left side of Table 1, descriptive statistics for the entire questionnaire survey indicated that skewness and kurtosis values were all within an acceptable range (i.e., limits of

**Table 1. Descriptive statistics of trait level indices for mind wandering in total and subsamples.**

| | Whole sample (N = 176) | | | Subsample (N = 104) | | | |
|---|---|---|---|---|---|---|---|
| **Measure** | **Mean (SD)** | **Skewness** | **Kurtosis** | **Mean (SD)** | **Skewness** | **Kurtosis** | **Cronbach's alpha** |
| **Inv-AS** | 28.26 (6.6) | −0.42 | −0.15 | 29.30 (6.0) | 0.48 | −0.10 | 0.88 |
| **DDFS** | 23.88 (9.2) | 0.19 | −0.39 | 23.10 (9.0) | 0.27 | −0.06 | 0.91 |
| **MWQ** | 18.02 (4.2) | -0.36 | −0.38 | 17.60 (4.2) | −0.56 | −0.34 | 0.61 |
| **IMI** | N/A | N/A | N/A | 29.06 (8.1) | 0.17 | 0.16 | 0.71 |
| **Probe** | N/A | N/A | N/A | 2.75 (1.0) | 0.49 | −0.36 | N/A |

Inv-AS, inverted scores of the apathy scale with higher scores indicating greater motivation; DDFS, daydream frequency scale; MWQ, mind wandering questionnaire; IMI, intrinsic motivation inventory; Probe, the rate of mind wandering measured by experience sampling.

±2) [23]. The results of the SART were as follows: mean Probe was 2.75 (SD: 1.0, range: 1.1–5.6), mean reaction time was 369ms (SD: 55, range: 281–569), and mean true negative rate (i.e., rate of correct [no-go] response to the target) was 0.71 (SD: 0.17, range: 0.17–1). Based on the reaction time and true negative rate, three participants who produced an outlier (a value $3^*$SD from the mean) were excluded from the state level analysis.

## Trait level association between motivation and mind wandering

The scatter plots and results of the correlation analyses for the trait level association between motivation and MW are described in the left half of Fig 2A. The Inv-AS was significantly associated with both MWQ, r = −0.26, p = 0.001, and DDFS, r = −0.26, p = 0.001, indicating that higher trait motivation was related to lower trait MW. The correlation between MWQ and DDFS was also confirmed, r = 0.52, p < 0.001. When we controlled for DDFS in the relationship between Inv-AS and MWQ, we found a significant partial correlation ($r_{partial}$ = −0.17, p = 0.024). Further, when the MWQ was controlled for, a significant partial correlation between Inv-AS and DDFS was found ($r_{partial}$ = −0.16, p = 0.030).

Approximately two-thirds of the participants took part in the following state level analysis. As presented in the right side of Table 1, the distribution parameters were acceptable in the subsample and total sample. We conducted Welch's t-test for each index. Even without considering statistical multiplicity, there was no difference between the two samples, $ps$ > 0.163, indicating that a selection effect was not found. Next, we repeated the correlation analysis for this subsample. A significant trait level association between motivation and MW was also

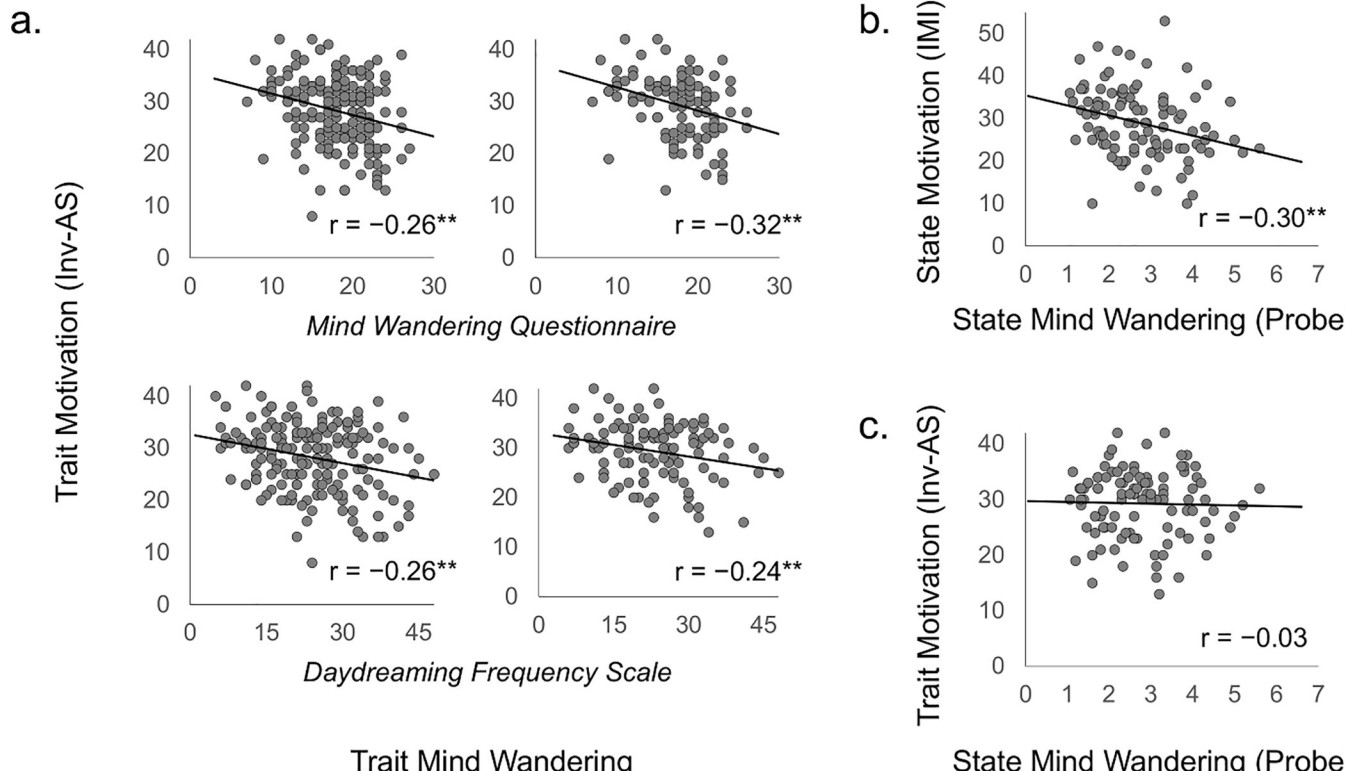

**Fig 2.** The associations between motivation and mind wandering at the trait level with total sample (n = 176; left half, a) and selected sample (*n* = 104; right half, a), and at state level with the selected sample (*n* = 104; b), and associations between trait motivation and state mind wandering (*n* = 104; c). Inv-AS, inverted scores of the apathy scale with higher scores indicating greater motivation; IMI, intrinsic motivation inventory, p < 0.05*, p < 0.01**.

confirmed in the subsample (Inv-AS correlation with MWQ was $r = -0.32$, $p < 0.001$, and with DDFS was $r = -0.24$, $p = 0.012$; right half in Fig 2A). The partial correlation analyses for the difference between MWQ and DDFS in this selected sample indicated that only the association between AS and MWQ was significant, $r_{partial} = -0.23$, p = 0.011, after controlling for another MW index (InvAS-DDFS controlling for MWQ, $r_{partial} = -0.09$, p = 0.353).

### State level association between motivation and mind wandering

A state level correlation analysis was performed including the indices of motivation (IMI) and MW (Probe). Those descriptive data are also shown in the right side of Table 1. A significant correlation between IMI and Probe, $r = -0.30$, $p = 0.002$, (Fig 2B) was found, indicating that higher state motivation was related to lower occurrence of MW during the SART. Moreover, this relationship remained significant when the task performance (i.e., true negative rate) was controlled ($r = -0.27$, $p = 0.005$).

### Relationship between trait and state level indices

We found a significant correlation between the trait and state level of motivation indices (i.e., AS and IMI), $r = 0.31$, $p = 0.001$. The trait-state association in MW was not consistent between measures in which the state MW's correlation with MWQ was significant, r = 0.23, p = 0.017, but the correlation with DDFS was not significant, $r = -0.13$, $p = 0.173$. This dissociation will be discussed later. Given the above findings, one can speculate that trait motivation influences the level of state MW. A scatter plot and correlation coefficients are depicted in Fig 2C, showing no correlation, $r = -0.03$, $p = 0.792$. To investigate which factors had an impact on state MW, we conducted a multiple regression analysis to predict state MW (Probe) based on their trait and state motivation (Inv-AS and IMI), and indices of trait MW (MWQ and DDFS) with a stepwise method (Criteria: probability-of-F-to-enter $< = 0.05$, probability-of-F-to-remove $> = 0.10$). A significant regression equation was found, $F(1,102) = 9.73$, p = 0.002, with the predictor of IMI, $b = -0.30$, $p = 0.002$, but the contribution was low, $R^2_{adj} = 0.08$. Neither the trait measures of MW nor the trait motivation was significant predictors.

## Discussion

State motivation refers to the present motivation to engage and persist due to an inherent interest and pleasure associated with the activity at hand, while trait motivation is a stable and enduring disposition, affected by individual characteristics such as personality [12,24]. Intuitively, there should be an association between motivation toward a task and MW while executing that task, which has been confirmed in prior work [6,25]. We replicated this state level association independent of task performance and demonstrated that such an association was also present at the trait level.

In this study, the trait motivation was assessed via AS. Although this scale was originally developed for clinical populations, the scores ranged from 0 to 29 in our healthy sample, indicating that it had sufficient dispersion to assess its association with other factors. It also shows that relative apathetic characteristics could be observable in healthy populations as reported in previous studies [10,11]. For trait MW, we used DDFS and MWQ. Owing to their significant association [16], in addition to the reported validity [17], both can be assumed to have assessed individual MW tendency as a trait characteristic. In the whole sample, both indices were found to have significant associations with trait motivation independent of each other. As for the selected sample; however, an independent association was confirmed only in MWQ. Additionally, the trait-state significant association in MW was only confirmed in the MWQ measure. Although we aimed to show the significant associations in both measures of both

samples, the current results could be expected based on previous findings wherein the MWQ showed higher sensitivity when representing a tendency toward MW than DDFS [26]. Day-dreaming, a core concept in DDFS, refers to a stimulus-independent thought that does not occur during a primary task, while MW involves a redirection of attention away from the task. In other words, the DDFS might tap intentional MW while the MWQ may include "unwanted" or "unintentional" MW. Since motivation relates differently to intentional and unintentional MW [6,27], future research should assess MW intentionality to understand its relationship with motivation. Nevertheless, the results indicated a significant trait level association between motivation and MW.

Attempts to understand the phasic character of MW might be fruitful. As demonstrated in previous studies [6,25], we have shown that less motivation toward a task leads to more MW during the task. The correlation between trait and state MW was not substantial ($r = 0.23$), being similar in size to that of a previous report [13]. This indicates that state MW is a phasic phenomenon driven by a range of factors, one being state motivation. Although we expected that the trait level motivation would affect state level MW, mediated by the trait level MW and/or state level motivation, we did not find such a relationship. The possibility that the association was indirect, such as via a mediation effect, can be rejected because of the null direct effect. Although the lack of a direct relationship between the independent variable (X) and dependent variable (Y) in mediation analysis can occur for various reasons, it is mainly due to the "competitive mediation" in which the indirect and direct paths represent opposite signs [28,29]. In the current case, those paths must, theoretically, represent the same signs, making it necessary to confirm the direct associations to analyze the mediation. Since our data did not satisfy this assumption, in addition to the results of multiple regression analysis, we conclude that, at least from the current dataset, there is no association between trait motivation and state MW. Only the state motivation factor affects current MW among the measurements in this study.

Finally, there were some limitations to this study. First, the surveys conducted for the state level investigation were unbalanced. MW was proved periodically throughout the task, while motivation was assessed only once. Although this point can be problematized for validity of the results, we did not believe that this was a critical problem considering that state motivation is assumed to be stable across, at least, several minutes [25], and that our pilot study found a strong correlation ($r > 0.8$) between the motivations before and after the SART. Additionally, the SART might not be a pure measurement of state MW; although, state MW was operationally defined by SART in the current study. However, so far at least, the probe caught method is a popular and reliable method to measure MW [30]. Second, we did not distinguish the content of MW in the SART, including different forms of distraction which might be important for assessing MW [30,31]. Although such discrimination would be desirable to assess state MW, the current study might not be the appropriate case because the trait indices used here also do not distinguish the content of MW. Third, although we have denoted the good dispersion of AS scores, it seems that the AS did not capture the upper level of motivations rigorously (i.e., smaller dispersions in higher motivation). Since this might affect the observation, it is recommended that subsequent studies should use different scale for assessing motivation which could capture a wider range of motivational traits.

## Conclusion

We empirically demonstrated that motivation and MW are generally associated at both trait and state levels. Motivation is one of the predictors for the occurrence of MW at both levels. As far as the authors are aware, no study had shown the trait level association. Regarding

causality, we currently acknowledge that motivation affects the rate of MW at the state level [6]; however, the opposite direction might be possible. For example, the person who recognizes more MW thinks they lack motivation to perform a task and we do not understand the causality at the trait (or daily life) level. In addition, the results indicated a dissociation along the trait-state dimension, which suggest that the state MW would be more phasic than expected as found in a previous study [13] and that a different mechanism might cause the relationship between motivation and MW. Although an intuitive perspective can afford the idea that less motivation to perform the ongoing task causes more MW, it seems unlikely that apathetic participants experience MW habitually, that is, in every moment or state of life. More statistically powerful assessment would be needed for this point. Thus, future MW studies should assess motivation as a factor for MW, paying attention to its trait or state dimension.

## Author Contributions

**Conceptualization:** Toshikazu Kawagoe.

**Data curation:** Toshikazu Kawagoe.

**Formal analysis:** Toshikazu Kawagoe.

**Funding acquisition:** Toshikazu Kawagoe.

**Investigation:** Toshikazu Kawagoe.

**Methodology:** Toshikazu Kawagoe.

**Resources:** Toshikazu Kawagoe.

**Writing – original draft:** Toshikazu Kawagoe.

**Writing – review & editing:** Keiichi Onoda, Shuhei Yamaguchi.

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
