## [Decision Letter · Decision Letter 0]

1 Jun 2020

PONE-D-20-10065

The Association of Motivation with Mind Wandering in Trait and State Levels

PLOS ONE

Dear Dr. Kawagoe,

Thank you for submitting your manuscript to PLOS ONE. I have invited 2 experts on research on mind-wandering and the SART to evaluate your manuscript, and I have read the paper myself. I would like to thank the reviewers for the careful and thoughtful comments. You can find the reviewer's comments appended below.

Overall, both reviewers believe that you present an interesting set of data that can make a fine contribution to the literature after revision. I agree with their assessment; therefore I am inviting a MAJOR REVISION. All of the reviewer's comments are amenable to be addressed in a revised version of the manuscript. I will not reiterate all the points made by the reviewers: they span providing a better argumentation in the introduction for associations between stait and trait assessments; including more details regarding the scales you used (e.g., example items); and clarifying points regarding the methods used (e.g. regression vs. mixed-effects model). Please include a detailed response letter in which you address each of their comments and indicate how you have changed the manuscript in accordance with it (or in case you do not agree with their point, please clarify your reasons). Also please mark in the revised manuscript all the changes that were implemented (e.g., by having modifications presented in a different color) in order to facilitate a new round of reviews. If you decide to resubmit your paper to PLOS ONE, I will invite one or both of the reviewers to evaluate again your submission. Hence please try your best to address all of their concerns.

In addition to the reviewer's comments, I would like to include a few points for addressing in a revision:

(1) Please indicate the full contact information of the ethics committee from which the data could be requested. Please also indicate the ethical reasons that prevent public sharing of the data.

(2) p. 6, please define what is a sizeable correlation: provide an actual number. You can even drop the term sizeable, the number can speak for itself.

(3) p. 9, please define what a true negative rate is.

(4) p.9, please list the reliability of each scale in Table 1.

(5) Similarly to Reviewer 1, I think you have a rich data-set with the SART, and I would like to see a detailed assessment of how performance in the SART relate to self-reported mind-wandering. One can evaluate, for example, whether mean and variability in RTs predict MW and accuracy in no-go trials. If probes followed no-go trials, one can assess whether no-go accuracy is associated with reported MW.

(6) At moments, I felt the paper had too many abbreviations. Consider using words instead of abbreviations because translating their meaning creates a burden to the reader.

Thank you for considering PLOS ONE as an outlet for your research.

We look forward to receiving your revised manuscript.

Kind regards,

Alessandra S. Souza, Ph.D.

Academic Editor

PLOS ONE

Journal Requirements:

Reviewers' comments:

Reviewer's Responses to Questions

**Comments to the Author**

1. Is the manuscript technically sound, and do the data support the conclusions?

Reviewer #1: Yes

Reviewer #2: Yes

2. Has the statistical analysis been performed appropriately and rigorously? 

Reviewer #1: No

Reviewer #2: Yes

3. Have the authors made all data underlying the findings in their manuscript fully available?

Reviewer #1: No

Reviewer #2: No

4. Is the manuscript presented in an intelligible fashion and written in standard English?

Reviewer #1: Yes

Reviewer #2: Yes

5. Review Comments to the Author

Reviewer #1: Summary of the research

This manuscript reports interesting research on the subject of Mind Wandering and its association with trait and state levels of motivation. The trait level of motivation was measured with a questionnaire designed for diagnosing clinical levels of apathy, which in inverse scoring represents one’s trait general levels of motivation, and state level with Intrinsic Motivation Inventory, which is designed for measuring motivation in schizophrenic patients. Mind Wandering was assessed with established scales such as the Mind Wandering Questionnaire (Mrazek et al, 2013), and the Daydreaming Frequency Scale (Giambra, 1993). State Mind Wandering was assessed with repeated experience sampling in a SART task and state motivation was measured once at the end of the SART task, with the Intrinsic Motivation Inventory.

The results showed that, replicating previous research, there were significant correlations between the trait levels of Mind Wandering and motivation. However, state-level Mind Wandering was not predicted by any measures except the state level motivation, but the variance explained by this predictor was low (R2adj = 0.08). However, the regression analysis conducted with this data did not take into account that the assumption of non-independence of the data points (as it is within-subjects’ measurement) is not met. My recommendation is to recompute the regression analysis with a linear mixed model, which is specifically developed for within-subjects’ designs and accounts for non-independence of within-person measurements. This review also highlights several other smaller concerns, which cannot be addressed directly, but may be important to acknowledge or discuss in the manuscript.

In summary, this research is an interesting contribution towards the study of Mind Wandering, written in concise format, in good English with an agreeable sense of flow, and will be of interest to a broad audience.

Major point 1 – non-independence of observations in the data submitted to the regression analysis. Recommendation: linear mixed model

On Page 12, the authors report: «To investigate which factors had an impact on state MW, we conducted a multiple regression analysis to predict state MW (Probe) based on their trait and state motivation (Inv-AS and IMI), and indices of trait MW (MWQ and DDFS) with a stepwise method...”

This regression analysis should be conducted including subject (i.e., participant) as a random effect along with estimation of the fixed effects (i.e., the contribution of the measured variables). Because the design of this subsample is within-subjects (i.e., each participant contributed to all measurements), the data within people are more similar than data between people, and thus the assumption of independence of observations, which is an assumption of linear regression, is not met. Making this explicit in the model is necessary to estimate fixed effects (i.e., the contributions of the trait and state measurements) adequately. E.g., because the contributions of the individuals from different scales are more similar within the individual, the model can overestimate the strength of the statistical relationship, if within-participants’ data dependency is not specified in the model.

Accordingly, for this analysis linear mixed effects models are the appropriate tool (Singmann, H., & Kellen, D. (2017). An introduction to mixed models for experimental psychology. New methods in neuroscience and cognitive psychology). Using p-value statistics, this is analysis can be conducted with the free software R using the package “lme4” (Bates, Mächler, Bolker, & Walker, 2015) or package “afex” (Singmann, Bolker, Westfall, & Aust, 2016).

Major point 2 – The authors conclude that “there is no association between trait motivation and state MW” (page 15). This conclusion may be problematic, seeing as it is based on a statistical null result. Seeing as p-value testing can only reject H1 but not prove H0, this statement could be reformulated. It could also be specified whether the authors conclude that this association does not exist at all or do they wish to express that it does not exist in their data.

Minor points

1) It would be helpful to the reader to provide more detailed descriptions of the questionnaires (i.e., how many items they contain, what the possible range of scores is).

2) A possible concern with the Apathy Scale (which could be discussed) is that the scale only goes from 0 to minus X points (as I assume; please see the point above about the interest to describe the scales more thoroughly). I.e., if I correctly understand the scale idea, it is conceived to measure apathy on a continuum from 0 apathy to very high level of apathy. However, if so, this way of measurement does not capture the upper region of motivation, namely, higher motivation than the average. It seems likely, therefore, that the AS scale can capture the area of average to very low motivation, whereas it cannot capture motivation levels that are higher than average. Is this true, and if yes, does this have implications for the power of the data to observe the hypothesized relationship with Mind Wandering?

3) On page 12, section “Relationship between trait and state level indices», reports: “We found a correlation between the trait and state level of each motivation index, r = 0.31, p = 0.001. After several times of reading the manuscript, I now assume that this statement refers to “We found significant correlation between the IMI and the AS”. If my understanding is correct, it would be helpful to the reader to reformulate the sentence. At the present formulation it makes the impression that there were more than two indices, and they were correlated multiple times, which is probably not what the authors intended.

Very small points

1) On Page 3, the authors write «MW is not the umbrella term for the psychological phenomenon to which we refer, including task-unrelated thought, stimulus-independent thought, self-generative thought, and zoning/tuning out.» Considering the context, I understand they wish to say MW IS the umbrella term?

2) The publication of Smallwood and Schooler (2006 ,The Restless Mind) is not a study, but a review.

3) Page 7, «Behavioral Task and Questionnaire for State Level Investigation». Please cite the developers of the SART task (Robertson, I. H., Manly, T., Andrade, J., Baddeley, B. T., & Yiend, J. (1997). Oops!’: Performance correlates of everyday attentional failures in traumatic brain injured and normal subjects. Neuropsychologia, 35(6), 747–758).

Reviewer #2: Summary

The authors investigate the relationship between mind wandering and motivation on a state and on a trait level. To this end, they measure individual differences in trait mind wandering with two questionnaires and individual differences in (general) motivation with one questionnaire. State mind wandering is assessed during a continuous response task via thought probes which ask participants from time to time to indicate whether they were on-task or off-task at this very moment. Motivation to perform this particular task was assessed after the task with another questionnaire. Results show relations between mind wandering and motivation on both the state and the trait level. However, trait motivation was not related to state mind wandering.

Evaluation

This manuscript addresses an interesting and timely topic namely the relationship between mind wandering and motivation. The reported results are interesting but the authors could provide some more information that would help to better evaluate the present results.

1) Building on Seli et al. (2016, 2019) the authors argue that they assess state mind wandering by including thought probes in an ongoing task. I am not completely convinced that this is indeed a pure state measure of mind wandering. In line with my concern, the authors report correlations of .5 between the state and trait mind wandering measures. Seli et al. may provide good arguments for their view but the authors currently do not make them transparent. In order to convince the readers that state and trait mind wandering is dissociable the authors should elaborate their and Seli et al.’s rationale in the Introduction section.

2) The authors should provide example items for all questionnaires they used so that the reader can better understand how they assess the constructs of interest.

3) Relatedly, it would be important to know whether there is a conceptual overlap between apathy and trait mind wandering. That is, are the items used to assess the one construct semantically similar to the items used to assess the other one or not?

4) The authors should indicate how they determined the thought probe presentation within the SART. It would be particularly interesting to know whether they occurred primarily after go or after no-go trials.

5) If the thought probes were presented after no-go trials mostly, it might be the case that participants used their SART performance for evaluating whether they had been on-task or off-task. Similarly, it might be that they used their SART performance to evaluate their motivation to perform this task after task completion. For this reason, the authors should also calculate and report SART error rates and correlate them with state mind wandering and state motivation. My question would be whether the correlation between state mind wandering and state motivation is still present when task performance is controlled for.

6) It would be really nice if the authors were able to make this data set available. Currently, the authors rather generically state that they refrain from doing so because of ethical issues. But what are the exact ethical concerns preventing the authors from sharing their data publically?

6. PLOS authors have the option to publish the peer review history of their article (what does this mean?). If published, this will include your full peer review and any attached files.

Reviewer #1: Yes: Andra Arnicane

Reviewer #2: No

---

## [Decision Letter · Decision Letter 1]

28 Jul 2020

The Association of Motivation with Mind Wandering in Trait and State Levels

PONE-D-20-10065R1

Dear Dr. Kawagoe,

We’re pleased to inform you that your manuscript has been judged scientifically suitable for publication by both reviewers. Thank you for being responsive to the comments raised during the review process. Your paper will be formally accepted for publication once it meets all outstanding technical requirements.

Kind regards,

Alessandra S. Souza, Ph.D.

Academic Editor

PLOS ONE

Additional Editor Comments (optional):

Reviewers' comments:

Reviewer's Responses to Questions

**Comments to the Author**

1. If the authors have adequately addressed your comments raised in a previous round of review and you feel that this manuscript is now acceptable for publication, you may indicate that here to bypass the “Comments to the Author” section, enter your conflict of interest statement in the “Confidential to Editor” section, and submit your "Accept" recommendation.

Reviewer #1: All comments have been addressed

Reviewer #2: All comments have been addressed

2. Is the manuscript technically sound, and do the data support the conclusions?

Reviewer #1: Yes

Reviewer #2: Yes

3. Has the statistical analysis been performed appropriately and rigorously? 

Reviewer #1: Yes

Reviewer #2: Yes

4. Have the authors made all data underlying the findings in their manuscript fully available?

Reviewer #1: No

Reviewer #2: No

5. Is the manuscript presented in an intelligible fashion and written in standard English?

Reviewer #1: Yes

Reviewer #2: Yes

6. Review Comments to the Author

Reviewer #1: The review comments have been adequately addressed and the authors' approach has been clarified in the rebuttal letter. The manuscript has benefitted from the review and the message now comes across more clearly.

Reviewer #2: I believe the authors could have been a bit more responsive (e.g., reporting SART performance data). However, the concerns I raised have been addressed appropriately.

7. PLOS authors have the option to publish the peer review history of their article (what does this mean?). If published, this will include your full peer review and any attached files.

Reviewer #1: No

Reviewer #2: No

---

## [Editor Report · Acceptance letter]

3 Aug 2020

PONE-D-20-10065R1 

The Association of Motivation with Mind Wandering in Trait and State Levels 

Dear Dr. Kawagoe:

I'm pleased to inform you that your manuscript has been deemed suitable for publication in PLOS ONE. Congratulations! Your manuscript is now with our production department. 

Kind regards, 

on behalf of

Dr. Alessandra S. Souza 

Academic Editor

PLOS ONE